# Effects of Plant Growth Promoting Rhizobacteria on the Content of Abscisic Acid and Salt Resistance of Wheat Plants

**DOI:** 10.3390/plants9111429

**Published:** 2020-10-24

**Authors:** Tatiana Arkhipova, Elena Martynenko, Guzel Sharipova, Ludmila Kuzmina, Igor Ivanov, Margarita Garipova, Guzel Kudoyarova

**Affiliations:** 1Ufa Institute of Biology, Ufa Federal Research Centre, Russian Academy of Sciences, Prospekt Oktyabrya 69, 450054 Ufa, Russia; tnarkhipova@mail.ru (T.A.); evmart08@mail.ru (E.M.); g.v.sharipova@mail.ru (G.S.); ljkuz@anrb.ru (L.K.); i_ivanov@anrb.ru (I.I.); 2Department of Biology, Bashkir State University, Zaki Validi st. 32, 450076 Ufa, Russia; marmaritag@list.ru

**Keywords:** salinity, abscisic acid, hydraulic conductivity, plant growth-promoting bacteria

## Abstract

Although salinity inhibits plant growth, application of appropriate rhizosphere bacteria can diminish this negative effect. We studied one possible mechanism that may underlie this beneficial response. Wheat plants were inoculated with *Bacillus subtilis* IB-22 and *Pseudomonas mandelii* IB-Ki14 and their consequences for growth, water relations, and concentrations of the hormone abscisic acid (ABA) were followed in the presence of soil salinity. Salinity alone increased ABA concentration in wheat leaves and roots and this was associated with decreased stomatal conductance, but also with chlorophyll loss. Bacterial treatment raised ABA concentrations in roots, suppressed accumulation of leaf ABA, decreased chlorophyll loss, and promoted leaf area and transpiration. However, water balance was maintained due to increased water uptake by inoculated plants, brought about in part by a larger root system. The effect may be the outcome of ABA action since the hormone is known to maintain root extension in stressed plants. Root ABA concentration was highest in salt-stressed plants inoculated with *B. subtilis* and this contributed to greater root hydraulic conductivity. We conclude that bacteria can raise salt resistance in wheat by increasing root ABA, resulting in larger root systems that can also possess enhanced hydraulic conductivity thereby supporting better-hydrated leaves.

## 1. Introduction

High salt concentration in the soil solution is widespread, and the area of saline land makes up to 6% of arable land. Salinity negatively affects plant growth and development by reducing the availability of water. This ‘osmotic component’ of salt stress [1] reduces tissue hydration, thereby inhibiting cell expansion growth and resulting in stomatal closure, photosynthesis inhibition, oxidative stress, and accelerated senescence [2]. Along with the osmotic component, the negative effect of salinity is due to the toxicity of ions (especially sodium), which gradually penetrate into the plant tissues [3]. In general, the negative effect of salinity results in a decline in plant productivity and dictates the need to find ways for increasing their salt-resistance. One way to solve this problem is to use growth stimulating bacteria. Plant inoculation with growth stimulating bacteria has been shown to increase plant resistance to stressful influences ([4] and references therein). In the case of salinity, the protective action of bacteria is due to an increase in the activity of the antioxidant system [5], accumulation of osmotically active substances and osmoprotectors [6], maintenance of ionic homeostasis [7], and direct action of bacteria on plant growth explained by their ability to produce growth stimulating plant hormones [4]. Along with these mechanisms, the bacteria-induced increase in salt tolerance is associated with their influence on plant water relations manifested in changes in the rate of transpiration [8]. Nevertheless, information about the influence of bacteria on the rate of transpiration is contradictory. Thus, it was shown that stomatal conductance was decreased by bacterial inoculation, which helped to save water and maintain tissue hydration under conditions of water deficit [9]. This mechanism is especially important at the beginning of salinity action [10]. At the same time, an opposite effect of bacteria is described, i.e., an increase in stomatal conductivity and transpiration under bacterial influence against the background of salinity [11]. The advantage of this reaction is in that stomatal opening normalizes gas exchange and photosynthesis of plants, while transpiration flow provides the uptake of essential nutrients. However, this reaction leads to an increase in transpiration losses, which can negatively affect plant hydration. Surprisingly, bacteria-induced increase in stomatal conductivity and transpiration did not lead to a decrease in the plant tissue hydration suggesting that bacteria increased the flow of water from the roots [12]. Indeed, in some studies, an increase in hydraulic conductivity was detected under the influence of bacteria, which was associated with an increase in the expression of genes encoding membrane water channels aquaporins [13,14]. These few studies showing the ability of bacteria to increase hydraulic conductivity are widely cited in reviews [7,12,15], indicating the importance of this reaction for plant adaptation to water scarcity. However, the effect of bacteria on plant hydraulic conductivity remains poorly understood, and it is still a mystery how bacteria can affect hydraulic conductivity.

Abscisic acid (ABA) is well known to regulate water relations [16]. It was also shown that bacteria can affect the level of ABA in plants, but changes in concentration of this hormone are mainly interpreted as a mechanism controlling stomatal conductance [7,17]. However, this hormone can influence many other processes that occur in plants during salt stress. It is known that ABA is capable of regulating plant senescence [18]. This hormone can also affect hydraulic conductivity [19], but the possible role of ABA in regulating hydraulic conductivity has not been studied in plants inoculated with bacteria. To reveal involvement of ABA in bacteria-induced changes in hydraulic conductivity, it is important to identify changes in the concentration of this hormone in the roots of plants, since it is in the roots that the influence of ABA on hydraulic conductivity is most obvious [20,21]. However, ABA concentration in plants inoculated with bacteria was mainly studied not in the roots but in the leaves. These considerations determined the goal of our work, which was to identify the possible role of ABA in the action of bacteria on plant hydration, with an emphasis on changes in hydraulic conductivity, and to evaluate the significance of these reactions, along with other mechanisms of bacterial action, for the increase in plant salt tolerance.

## 2. Results

### 2.1. Field Experiments

Year-to-year variation of wheat crop yield was rather great, which was likely to be due to unstable temperature conditions in this region. Productivity was about 1.5 times higher in 2016 than in 2018, which may be explained by the fact that average temperature in May–June was 1.5 times higher in 2017 (17 °C) than in 2018 (12 °C). In 2016, the effect of bacterization on wheat crop yield was studied without artificial salinization of the soil. Under these conditions, crop yield was increased 1.4 and 1.5 times by bacterization of seeds with *Pseudomonas mandelii* IB-Ki1*4* and *Bacillus subtilis* IB-22, respectively, compared to unbacterized plants (Table 1). Later on, salinization of the soil was applied to follow effects of PGPR on plant salt resistance.

In 2018, plant productivity was studied against the background of salinization of the soil with 10% NaCl. Salt stress led to a sharp 4-fold decrease in yield, and although the bacterization with *P. mandelii* IB-Ki14 and *B. subtilis* IB-22 increased yield by 13% and 30%, respectively, wheat productivity was still low. The association of bacteria and plants can be used to remediate soil with this level of salinity. However, from a commercial point of view, it is not sensible to grow plants under such conditions, and in 2019 the salinity level was lowered to 5%. Under these conditions, the yield decline was less sharp (two times lower than in the stress-free plants), and seed bacterization with *P. mandelii* IB-Ki14 and *B. subtilis* IB-22 led to a significant increase in plant productivity by 21% and 30%, respectively, compared to unbacterized plants. Comparison between the crop yield of wheat inoculated with either strain over all the years of field experiments showed that the yield increment was higher in the case of *B. subtilis* IB-22 (*p* ≤ 0.05, paired *t*-test). It was of interest to perform a more detailed study of the effects of bacterization on the morphometric and physiological plant indicators under the background of salinity, which was done in 2019.

*Morphometric and physiological characteristics in the field experiments*. Salinization caused a decrease in shoot mass and leaf area (Figure 1). The shoot fresh mass and leaf area of the stressed plants was about 50% of that of the control plants. In the salt-stressed plants, seed treatment with either *B. subtilis* IB-22 or *P. mandelii* IB-Ki14 significantly increased the mass of shoots and the leaf area by about 37% and 40%, respectively, compared to the unbacterized plants (Figure 1).

Measurement of RWC showed that salinity led to a decline in the hydration of the first leaf (from 83% to 79%) indicating an increase in the water deficit, which is equal to (100-RWC)% (Figure 2a). In salt-stressed plants, bacterization had a different effect on leaf hydration depending on the strain. Leaf RWC of plants treated with *B. subtilis* IB-22 did not differ from that of unbacterized salt-treated plants. However, leaves of plants treated with *P. mandelii* IB-Ki14 had lower RWC compared to either stressed unbacterized plants or those treated with *B. subtilis* IB-22.

Leaf osmolality was increased by salt stress (Figure 2b). Concentration of osmotics was higher in plants treated with *P. mandelii* IB-Ki14 than in salt-stressed unbacterized plants. In terms of osmolality, plants grown from seeds inoculated with *B. subtilis* IB-22 occupied an intermediate position between stressed unbacterized plants and those treated with *P. mandelii* IB-Ki14.

Salt stress resulted in ABA accumulation in both roots and shoots of wheat plants (Figure 3a). Inoculation with either of the bacterial strains significantly reduced the content of ABA in shoots compared to unbacterized plants subjected to salinity stress. Treatment with *B. subtilis* IB-22 led to 2-fold increase in the level of root ABA compared to unbacterized stressed plants, while in the case of *P. mandelii* Ki14, the level of ABA occupied an intermediate position between unbacterized stressed plants and plants treated with *B. subtilis* IB-22.

Chlorophyll concentration in the unbacterized plants was decreased by salinity (Figure 3b). This indicator was increased by *B. subtilis* IB-22 to the level of control (stress-free unbacterized plants that grew without NaCl), and plants treated with *P. mandelii* Ki14 had an intermediate content of chlorophyll between non-bacterized stressed plants and those inoculated with of *B. subtilis* IB-22.

Stomatal conductance was decreased by salinity (from 400 to 280 mmol m^−2^ s^−1^), while bacterial inoculation of plants had no significant effect on this parameter.

### 2.2. Laboratory Experiments

Analysis of variance of both fresh and dry mass of roots and shoots of wheat plants (Table 2) gave similar results: they showed significant effects of both inoculation and salt-treatment on the shoot mass, while only the effect of salt-treatment was significant in the roots. Nevertheless, the effect of salt-treatment on the root mass depended on inoculation supported by significant interaction between the factors (inoculation × salt treatment).

Salt-stress led to about 35% decline in the accumulation of the both fresh and dry mass of shoots and leaf area compared to the stress-free plants (control) (Figure 4a and Table 3). Introduction of bacteria into the rhizosphere of salt-stressed plants increased the shoot mass and leaf area compared to plants grown in the presence of NaCl, but without bacterial inoculation. The increase in shoot mass and leaf area was significantly higher when plant rhizosphere was inoculated with *B. subtilis* IB-22 than with *P. mandelii* IB-Ki14. Not only was the growth of the shoot, but also its development inhibited by salinity. Thereby, there was a delay in the appearance of the third leaf in salt stressed plants, and only in plants whose rhizosphere was inoculated with *B. subtilis* IB-22, a third leaf appeared by the end of the experiment.

Salinity inhibited root growth by 25%, and, in the variant without inoculation, both fresh and dry mass were lower than in the control (stress-free plants) (Figure 4b and Table 3). In the presence of NaCl, rhizosphere inoculation with *B. subtilis* IB-22 resulted in 16% increase in the roots fresh mass, while the tendency of its increase induced by *P. mandelii* IB-Ki14 was statistically insignificant. The tendency of the increase in root dry mass resulting from inoculation with both strains was statistically insignificant (Table 3).

Relative water content (RWC) was significantly decreased by salinity from 96% to 91% (Figure 5a). The downward trend in RWC was more pronounced when rhizosphere was inoculated with *P. mandelii* IB-Ki14. Measurement of leaf water potential gave results similar to those obtained for RWC: salt stress caused a decline in water potential and its further trend to greater effect in the case of rhizosphere inoculation with *P. mandelii* IB-Ki14 (Figure 5b).

In the absence of bacterial treatment, salinity reduced the level of transpiration by about 1.5 times, which was likely to be due to reduced leaf area (Figure 4) and stomatal conductance (Table 4). Bacteria increased the level of transpiration, and it was higher in plants whose rhizosphere was inoculated with *B. subtilis* IB-22 compared to inoculation with *P. mandelii* IB-Ki14 (Table 4). In plants treated with *B. subtilis* IB-22 transpiration was at the control level (plants grown under stress-free conditions). Salinity caused a decline in the hydraulic conductivity of the roots. The only exception was plants inoculated with *B. subtilis* IB-22, in which hydraulic conductivity was at the level of control (stress-free plants untreated with bacteria).

Salinity resulted in a statistically significant, although not great, increase in the concentration of MDA in leaves (by 22%). Concentration of MDA was lowered by introduction of *B. subtilis* IB-22 into rhizosphere, while in the case of *P. mandelii* IB-Ki14, the tendency of decline was statistically insignificant (Figure 6).

Two-way ANOVA showed significant effects of both salt-treatment and inoculation on ABA content in either roots or shoots. The effects of salt treatment on ABA level depended on inoculation in accordance with significant interaction between the factors (inoculation × salt treatment) revealed with the help of two-way ANOVA (Table 5).

Evaluation of the effects of salinity and bacterial treatment on ABA concentration revealed accumulation of this hormone in shoots and tendency of reduction of its level under the influence of bacteria, which was statistically significant in the case of *B. subtilis* IB-22. In the roots, on the contrary, salinity-induced accumulation of ABA was elevated by introduction of *B. subtilis* IB-22 into rhizosphere (Figure 7).

Comparison of plant responses in the field and laboratory conditions revealed similarities between them. Although plants grown in laboratory and field experiments were sampled at different stages of development (the appearance of the third leaf in the case of laboratory experiments and the stage of tillering in the field experiments), the similarity of effects confirms that the effects persisted throughout development. This was manifested in NaCl-induced decline in the growth and tissue hydration, while bacterial treatment contributed to maintenance of growth, despite the increased level of transpiration losses. Furthermore, both in the field and laboratory experiments, bacterial treatment reduced the stress induced accumulation of ABA in the leaves and increased the level of ABA accumulation in the roots. Similarity of these responses allowed us to consider the results of field and laboratory experiments as complementary.

## 3. Discussion

Analysis of the results obtained in both field and laboratory experiments shows that salt stress led to a number of negative consequences for wheat plants: a decrease in the accumulation of shoot mass, reduced concentration of chlorophyll and oxidative stress, manifested in an increase in the MDA level. Ultimately, all these effects led to a drop in plant productivity (Table 1). Negative effects of salinity were mainly brought about by water shortage due to a decline in its availability to plants. Water deficit was manifested in a decrease in the RWC and water potential of plants (Figure 2a and Figure 5), and plants adaptation involved accumulation of osmotically active substances and stomatal closure (Figure 2b, Table 4). These adaptive responses were likely to be due to an increase in ABA levels in stressed plants (Figure 3a and Figure 7) caused by the ability of this “stress hormone” to close the stomata [22] and to promote accumulation of osmotically active substances [23]. However, the effect of an increased ABA concentration can be contradictory, and this hormone may have negative impact on plants along with a protective one. It has been shown that ABA can contribute to salinity-induced senescence manifested in a reduced chlorophyll concentration and inhibition of photosynthesis [18]. Consequently, the decline in the level of stress-induced accumulation of ABA detected in the shoots of inoculated plants could contribute to the activation of plant growth by maintaining chlorophyll level and photosynthesis. Dependence of ABA content in the shoots on inoculation was in accordance with significant interaction between the factors (inoculation × salt treatment) revealed with the help of two-way ANOVA (Table 5).

Interestingly, treatment with bacteria increased the leaf area of plants subjected to salt stress by 1.5 times as compared to non-inoculated plants (Figure 1 and Figure 4a). This bacteria-induced increase in the leaf area, along with the unchanged stomatal conductance, apparently raised transpiration losses. Laboratory experiments in which transpiration was directly measured confirmed an increase in the rate of evaporation of water by leaves under the influence of bacteria (Table 4). It is important that the acceleration of water loss by *B. subtilis* IB-22 was not accompanied by a decline in plant hydration as compared to non-inoculated stressed plants. These results suggest an increase in the ability for water uptake by plants under the influence of these bacteria [12]. The increased ability of the roots to absorb water should have been facilitated by the activation of root growth, which was recorded under the influence of bacteria (significant in the case of *Bacillus*, while for *Pseudomonas* only an increasing trend was observed) (Figure 4b). ABA is known to support root growth under stress conditions [24]. Consequently, the increase in the level of this hormone in plant roots detected in our experiments, which is most clearly manifested in plants inoculated with *B. subtilis* IB-22, could promote the activation of root growth, causing an increase in their ability for water uptake. Dependence of root ABA content on inoculation is in accordance with significant interaction between the factors (inoculation × salt-treatment) revealed with the help of two-way ANOVA (Table 5). In addition, it is known that ABA can increase hydraulic conductivity of roots by increasing abundance of membrane water channels, aquaporins [20,21,25]. Measurement of hydraulic conductivity in our laboratory experiments revealed an increase in this indicator by *B. subtilis* IB-22. It is important that the increase in hydraulic conductivity in plants treated with *B. subtilis* IB-22 was accompanied by and was likely to be the consequence of the highest ABA level in plant roots.

It was suggested that ABA released by microorganisms into the soil may contribute to an increase in the concentration of this hormone in the roots [26]. Addition of synthetic ABA into the nutrient solution increased its concentration in the roots [21]. Although bacteria used in the present experiments did not accumulate ABA in vitro, this may change after their introduction into soil. Still, ABA concentration in the shoots was decreased by bacterial inoculation suggesting that the changes in ABA concentration in the plants detected in the present experiments were not merely due to the uptake of bacterial ABA by the plants

An increase in hydraulic conductivity was not recorded in the case of *P. mandelii* IB-Ki14. Thereby an increase in transpiration by these bacteria not accompanied by increased hydraulic conductivity, led to a decline in the RWC and water potential of the leaves. Nevertheless, inoculation with these bacteria stimulated plant growth, which was obviously due to the accumulation of osmotically active substances. The level was highest in plants inoculated with *P. mandelii* IB-Ki14 enabling the maintenance of turgor and plant growth [27]. It has been suggested that accumulation of osmotically active substances under stress is due to inhibition of plant growth that reduces the use of assimilates for growth and thereby releases them for osmotic adjustment [12,28]. However, in our case, an increase in the level of osmotics by *P. mandelii* IB-Ki14 was accompanied by activation of plant growth, and these results do not allow attributing osmotic adjustment to growth inhibition.

Above, we have already cited literature data indicating the ability of ABA to promote the accumulation of osmotically active substances [23], which allows linking the salt-induced increase in the level of osmotics with the accumulation of ABA. However, this explanation does not fit the results of the study of *P. mandelii* IB-Ki14 action showing the highest level of osmotics and reduced level of ABA in inoculated plants compared to stressed unbacterized plants. Obviously, the accumulation of osmotics under the influence of bacteria was not associated with the changes in ABA level. Nevertheless, bacteria-induced decrease in the level of this hormone in the shoots and its accumulation in the roots of salt-stressed plants were apparently important for some aspects of plant growth promotion by bacteria. Thereby the bacteria-induced decline in leaf ABA level is likely to be implicated in maintaining the level of chlorophyll and leaf growth of inoculated plants, while the increase in root ABA could activate the growth of roots and result in the increase of plant hydraulic conductivity.

The impact of bacteria on the ABA content in shoots has been evaluated several times before [9,11]. Our results are consistent with studies that found a decrease in the level of this hormone in plants subjected to salt stress. [11]. At the same time, we showed for the first time that a decline in the level of ABA in the shoots of inoculated plants can be accompanied by the accumulation of this hormone in the roots, which is likely to increase their absorption capacity. The ability of bacteria to influence distribution of hormones in plants has been shown by the example of bacterial action on auxin carriers [29]. A simultaneous decrease in the level of ABA in shoots and its accumulation in roots under the influence of bacteria suggest possible bacterial effect on the transport of ABA from the shoot to the roots. ABA carriers have recently been identified [30]. In the future, it is necessary to study the possible influence of bacteria on activity of these carriers. Our results indicate the prospects for work in this direction.

## 4. Materials and Methods

We studied the effects of bacterial inoculation of durum wheat seeds (*Triticum durum* Desf., cv. Bashkirskaya 27) on plant growth and productivity under conditions of salinity.

**Bacterial strains and culture media**. Gram-positive aerobic cytokinin-producing bacterium *Bacillus subtilis* IB-22 (GenBank MT590663) [31] and Gram-negative auxin-producing bacterium *Pseudomonas mandelii* IB-Ki14 (All-Russian collection of microorganisms VCM B-3250) [32] from the collection of microorganisms of Ufa Institute of Biology, RAS, were used for seed bacterization. Both bacteria are moderate halophiles (5–7% NaCl) [32] and were characterized by very low level of ABA accumulation measured in the culture media before inoculation (1–2 ng/mL, not less than 300 times lower than IAA and cytokinin concentrations in corresponding cytokinin (*Bacillus subtilis* IB-22 [33]) and auxin (*Pseudomonas mandelii* IB-Ki14 [34]) producing strains). Bacterial inoculates for seeds treatments were obtained by cultivating *B. subtilis* IB-22 on K1G medium as described in [34], while King B medium [35] was used for cultivation of bacteria of *P. mandelii* IB-Ki14. Strains of microorganisms were cultured in Erlenmeyer flasks with the appropriate nutrient medium on a shaker (160 rpm): for 72 h at 37 °C—in the case of *B. subtilis* IB-22, and for 48 h at 28 °C—in the case *P. mandelii* IB-Ki14. The biomass was separated from the nutrient solution by centrifugation for 20 min at 4000 rpm and diluted with tap water to reach appropriate inoculate density (indicated below).

**Field experiments** were performed in 2016–2019. Seeds were sown at a depth of 5–6 cm. There were four 1.5 m^2^ plots in an experimental field (54°50′ N, 55°44′ E, 170 m a.s.l.). The soil was a leached Chernozem of South wooden steppes of Bashkortostan. The ploughing horizon of unfertilized soil contained 3.65% C_org_ (organic carbon), was characterized by slightly acidic soil solution, high content of absorbed bases with dominating calcium (350 mmol/kg of Ca^2+^, 120 mmol/kg of Mg^2+^), and moderate availability of mobile phosphorus and alkaline hydrolysable nitrogen. Crop yields were measured in 2016, 2018, and 2019. The rainfall was about 150 mm in April–June of all the seasons. The average temperature in May–June was highest in 2016 (17°C) and lowest in 2018 (12°C). Salinity was artificially created by adding 5% (in 2019) and 10% (in 2018) NaCl solution to the soil at the rate of 10 L m^−2^. Microbe biomass was separated by centrifugation during 20 min at 4000 rpm and diluted in tap water to yield inoculate density of 10^6^ colony forming units (CFU) per seed. Carboxymethylcellulose sodium salt (Na-CMC) was added during bacterization and control treatment. Unbacterized seeds treated with Na-CMC served as the control.

At the tillering stage (28 days after planting) we determined the shoots mass, leaf relative water content (RWC), osmotic potential, concentration of ABA in the leaves and roots, and chlorophyll content. Crop yield was estimated at the end of the growing season.

**Laboratory experiments**. To ensure proper drainage, a layer of gravel was placed at the bottom of 500 cm^3^ vessels. After installing a glass tube (for gas exchange), the vessels were filled with 0.45 kg of dry soil (agro-chernozem clay-illuvial, characterized by medium humus content (6.3%) and slightly acidic soil solution), containing 10% sand. Three days before the experiment, the soil in the vessels was moistened with either water or 100 mm NaCl solution to reach 100% of the total field capacity. Wheat seeds were sterilized by soaking in a solution of 96% ethanol:3% H_2_O_2_ (1:1, *v*/*v*) for 5 min and then repeatedly washed with distilled water and germinated for 24 h in darkness. Ten seedlings were placed in each vessel, and suspension of bacterial cells (1 mL (10^8^ CFU mL^−1^) per each seedling) was simultaneously added to the rhizosphere by applying to the surface of the soil around roots of each seedling. Tap water (1 mL per each seedling) was added to the plants untreated with bacteria. Since total volume of water in the pot was about 200 mL, addition of 10 mL (1 mL to each of 10 seedlings in the pots) of either tap water or bacterial cells separated from the nutrient solution by centrifugation did not significantly influence concentrations of mineral nutrients or NaCl. Only single bacterial inoculation was performed without refreshed bacterial inoculation into the rhizosphere, since the used bacteria have been previously shown to efficiently colonize roots and persist on their surface [34]. Diluted supernatant obtained after peletting bacterial cells as well as the clear nutrient media processed in the same way as bacterial preparation did not influence plants growth. Plants were grown at 24 °C, 420 µmol m^−2^ s^−1^ PAR irradiance, and 14-h photoperiod. Plants grown in the soil without introduced bacteria served as the control. Soil moisture was maintained at 70% of the total field capacity by watering the plants with distilled water. The amount of water needed for irrigation was calculated by daily weighing vessels with plants. Transpiration, relative water content in the mature (first) leaf (RWC), water potential of the soil and leaves, and hydraulic conductivity were evaluated on the 13th day after the beginning of the experiments (the moment of plant inoculation). Roots and shoots were sampled for ABA analysis on the 11th day after inoculation. Fourteen days after plant inoculation, the fresh mass of roots and shoots and the area of leaves were evaluated.

**Characteristics of water relations**. Samples for measurement of tissue osmotic potential were obtained after freezing and thawing of leaves and subsequent centrifugation. Leaf osmolality was measured with the freezing point depression osmometer (Osmomat 030, Gonotec, Berlin, Germany). Leaf and soil water potentials were measured using a psychrometer (PSYPRO, “Wescor”, Logan, UT, USA). To determine RWC, mature first leaves of 10 plants were weighed and immersed in distilled water with the base; a vessel was tightly closed to saturate the air with moisture, and placed in darkness at room temperature. After 24 h, the turgid weight (TW) was determined after blotting and the dry mass was determined after drying for 24 h at 80 °C. Fresh weight (FW), dry weight (DW), and TW were used to determine relative water content: RWC = (FW − DW)/(TW − DW). Transpiration was measured by the weight loss of pots with plants, in which soil was covered for 4 h with parafilm to prevent water evaporation. The hydraulic conductivity of water pathway from roots to leaves (L) was calculated, as described [36], using the formula: L = T/[(Ψ_s_ − Ψ_l_) f.wt.], where T is the transpiration, f.wt. is the root fresh weight, and Ψ_s_ and Ψ_l_ are the water potential of the soil and leaf, respectively.

**ABA assay**. Shoots and roots of four plants were sampled from different vessels (number of replicates, *n* = 6) (shoots of four plants and roots of 25 plants were sampled for each repetition (*n* = 6) in the field experiments). Before homogenization, roots were thoroughly washed from the soil. The hormones were extracted for 16 h with 80% ethanol in a ratio of 1:10. Then the alcohol extract separated by filtration was evaporated to an aqueous residue, from which the abscisic acid was extracted with diethyl ether as described [37]. In short ABA was partitioned with diethyl ether from the aqueous residue, after its dilution with distilled water and acidification with HCl to pH 2.5 (ratio of organic to aqueous phases being 3:1). Then, ABA was transferred from the organic phase into a solution of NaHCO_3_, (ratio of aqueous/organic phase being 1:3), reextracted from the acidified aqueous phase with diethyl ether, ABA quantitative determination was performed with ELISA using specific antibodies as described [38]. The reliability of the method was due to the specificity of the antibodies obtained against ABA and the use of a modified ABA extraction method based on reducing the volume of extractants at each stage of extraction and re-extraction, which allows efficient extraction of ABA while reducing the amount of extracted impurities. The sufficiency of ABA purification prior to immunoassay was proved by studying the chromatographic distribution of the immunoreactive material, which showed that the peak of immunoreactivity coincides only with the position of the internal ABA standard.

**Membrane lipid peroxidation** was determined as the amount of malondialdehyde (MDA), a product of lipid peroxidation [39].

**Leaf chlorophyll index** was measured by DUALEX SCIENTIFIC+ (FORCE-A, France).

**Leaf area** was measured by ImageJ software (National Institutes of Health, Bethesda, MD, USA).

Statistics. Data were expressed as means ± S.E., which were calculated in all treatments using MS Excel. Significant differences between means were analyzed by two-way analysis of variance (ANOVA) with bacterial and salt treatment as main factors, and a least significance difference (LSD) test to discriminate means.

## 5. Conclusions

Thus, we demonstrated the importance of the ability of plants to absorb water for their adaptation to salinity-caused water shortage. The ability of bacteria to increase ABA content in roots was shown for the first time, and this response can promote both the root growth and plant ability to conduct water. We detected differences in the mechanisms of action of distinct bacterial strains on plant–water relations. In plants inoculated with *B. subtilis* IB-22, water flow was facilitated by increased hydraulic conductivity. However, this was not the case for *P. mandelii* IB-Ki14, whose inoculation did not change the hydraulic conductivity of plants, while the decline in leaf water potential was compensated by accumulation of osmotics. Increasing hydraulic conductivity appears to be a more effective mechanism than osmotic adjustment for increasing salt resistance under moderate stress. This is indicated by the fact that plants inoculated with *B. subtilis* IB-22, in which the first of these mechanisms was implemented, were characterized by higher concentration of chlorophyll, reduced level of MDA accompanied by greater increase in the growth and productivity than in plants inoculated with *P. mandelii* IB-Ki14 implementing the second mechanism. Identification of mechanisms providing a variety of bacterial effects on water relations of plants should be the subject of further research. However, our results indicate the importance of bacterial regulation of water flow for the activation of plant growth under salinity, as well as the significant contribution of ABA accumulation in roots to these processes.

## Figures and Tables

**Figure 1 plants-09-01429-f001:**
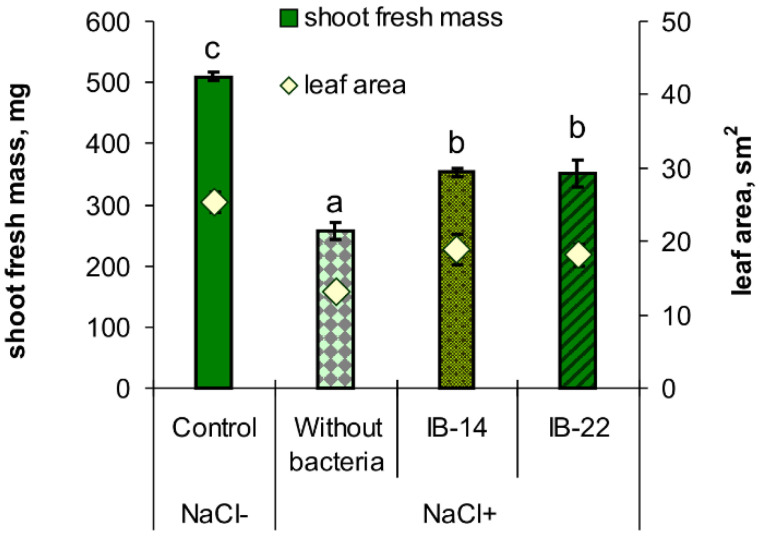
Effect of salt-stress and seed bacterial inoculation on leaf area and fresh shoot weigh of wheat plants at the stage of tillering in the field experiments: IB-14*—Pseudomonas mandelii* IB–Ki14; IB-22—*Bacillus subtilis* IB–22; control—option without bacterization and salinity. Significantly different means (*n* = 10) for each variable are labeled with different letters (*p* ≤ 0.05; LSD test).

**Figure 2 plants-09-01429-f002:**
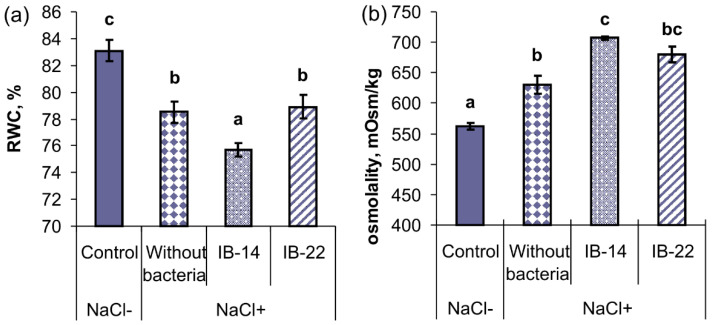
Effect of salt-stress and seed bacterial inoculation on leaf relative water content (**a**) and osmolality (**b**) of wheat plants at the stage of tillering in the field experiments: IB-14*—P. mandelii* IB–Ki14; IB-22—*B. subtilis* IB–22; control—option without bacterization and salinity. Significantly different means for each variable are labeled with different letters (*n* = 5, *p* ≤ 0.05; LSD test).

**Figure 3 plants-09-01429-f003:**
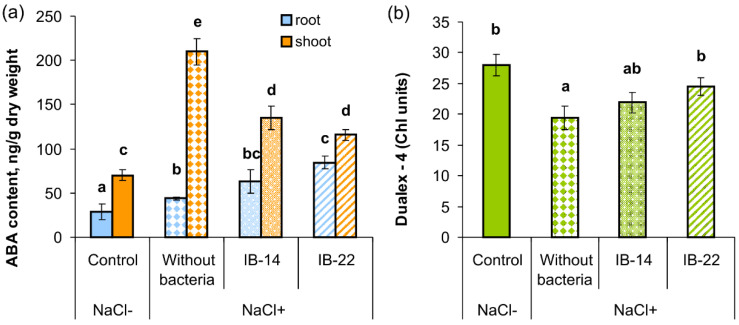
Effect of salt stress and seed bacterial inoculation on shoot and root ABA concentration (*n* = 6), (**a**) and leaf chlorophyll (Chl a + Chl b) concentrations in wheat plants (*n* = 25), (**b**) at the stage of tillering in the field experiments: IB-14*—P. mandelii* IB–Ki14; IB-22—*B. subtilis* IB–22; control—option without bacterization and salinity. Significantly different means for each variable are labeled with different letters (*p* ≤ 0.05; LSD test).

**Figure 4 plants-09-01429-f004:**
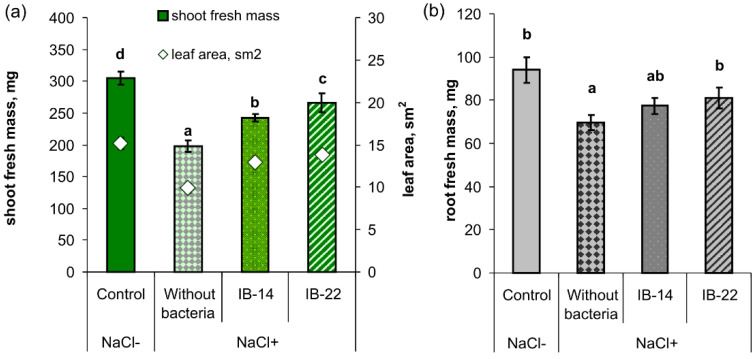
Effects of 100 mM NaCl and rhizospehere inoculation with *P. mandelii IB*-Ki14 (IB-14) and *B. subtilis IB*-22 (IB-22) on the shoot (*n* = 40) (**a**), root fresh mass (*n* = 8) (**b**), and leaf area (*n* = 20) (**a**) of 14-days-old wheat plants. Control—option without bacterization and salinity. Significantly different means for each variable are labeled with different letters (*p* ≤ 0.05; LSD test).

**Figure 5 plants-09-01429-f005:**
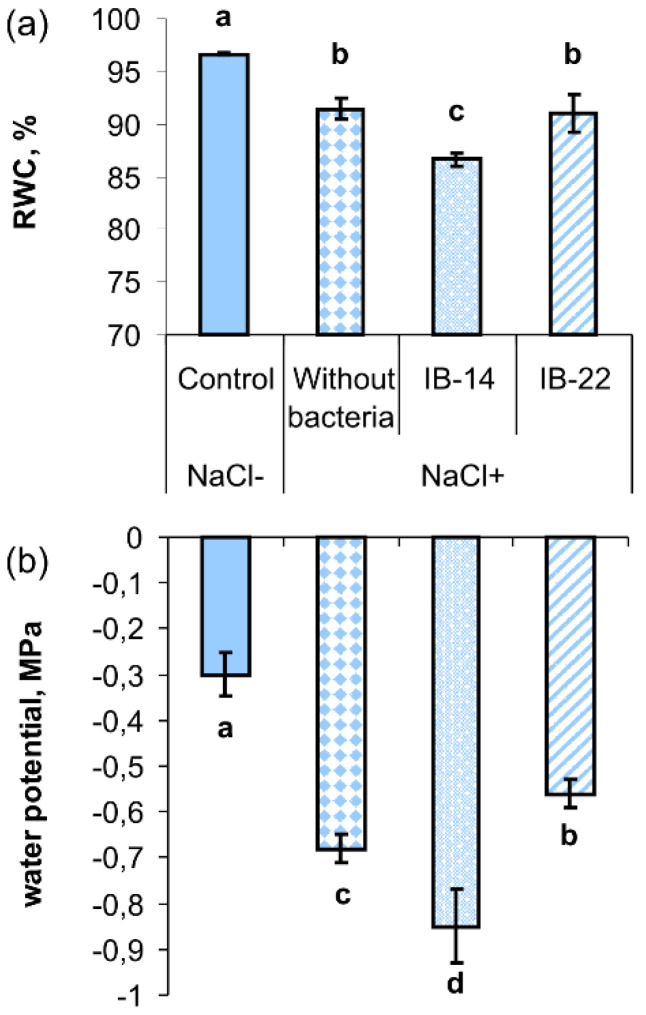
Effects of 100 mM NaCl and rhizospehere inoculation with *P. mandelii IB*-Ki14 (IB-14) and *B. subtilis IB*-22 (IB-22) on the relative water content (**a**), leaf water potential (**b**) of 13-days-old wheat plants. Control—option without bacterization and salinity. Significantly different means for each variable are labeled with different letters (*n* = 8, *p* ≤ 0.05; LSD test).

**Figure 6 plants-09-01429-f006:**
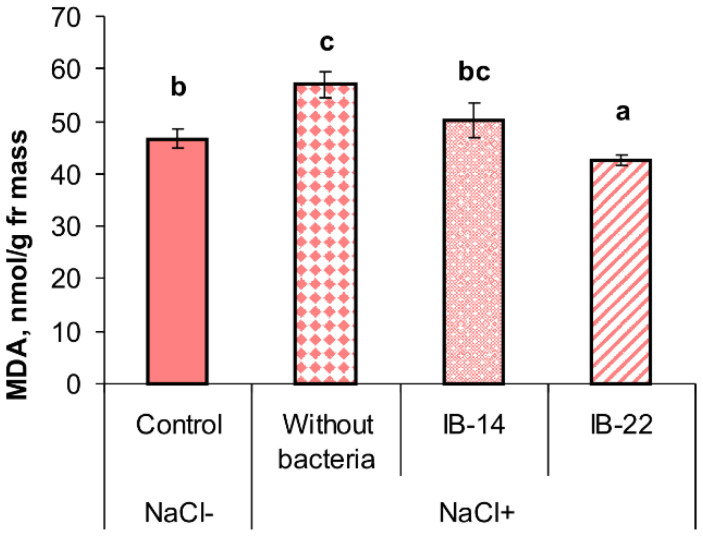
Effects of 100 mM NaCl and rhizospehere inoculation with *P. mandelii IB*-Ki14 (IB-14) and *B. subtilis IB*-22 (IB-22) on MDA content in 13-days-old wheat plants. Control—option without bacterization and salinity. Significantly different means for each variable are labeled with different letters (*n* = 6, *p* ≤ 0.05; LSD test).

**Figure 7 plants-09-01429-f007:**
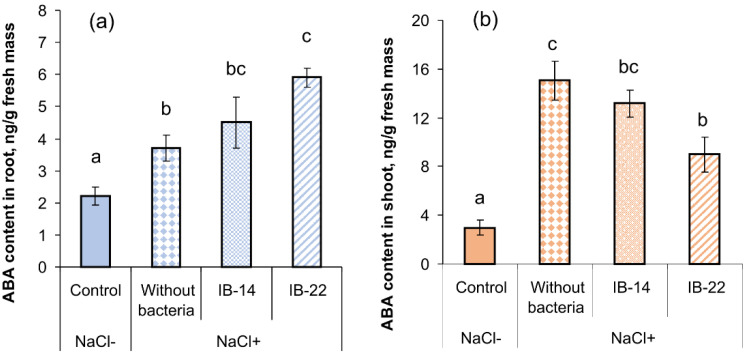
Effects of 100 mM NaCl and rhizospehere inoculation with *P. mandelii IB*-Ki14 (IB-14) and *B. subtilis IB*-22 (IB-22) on ABA concentration in roots (**a**) and shoots (**b**) of 11-days-old wheat plants. Control—option without bacterization and salinity. Significantly different means for each variable are labeled with different letters (*n* = 6, *p* ≤ 0.05; LSD test).

**Table 1 plants-09-01429-t001:** Effect of seed bacterization on crop yield under stress-free conditions and salinity.

Year of Testing	NaCl Concentration %	Crop Yield, g of Seeds m^2^		
		Without Bacterization	Seed Bacterization with	
			*P. mandelii* IB-Ki14	*B. subtilis* IB-22
2016	0	633 ± 12 ^a^	876 ± 8 ^b^	942 ± 10 ^c^
2018	0	400 ± 11 ^d^	557 ± 16 ^e^	581 ± 14 ^e^
	10	93 ± 4 ^a^	105 ± 4 ^b^	121 ± 4 ^c^
2019	0	430 ± 11 ^c^	478 ± 11 ^d^	540 ± 12 ^e^
	5	209 ± 5 ^a^	254 ± 6 ^b^	273 ± 6 ^c^

Significantly different means for each year are labelled with different letters (*n* = 600, *p* ≤ 0.05; LSD test).

**Table 2 plants-09-01429-t002:** Analysis of variance of fresh and dry mass of roots and shoots of wheat plants untreated with bacteria or inoculated with either *B. subtilis* IB-22 or *P. mandelii* IB-Ki14 (inoculation) grown in the presence and absence of NaCl (salt treatment). *p* values are presented for effects of inoculation and salt-treatment and interaction of these factors (NS—non-significant).

Effect or Interaction	Root		Shoot	
	Fresh Mass	Dry Mass	Fresh Mass	Dry Mass
Salt treatment	*p* ≤ 0.05	*p* ≤ 0.01	*p* ≤ 0.000001	*p* ≤ 0.000001
Inoculation	NS	NS	*p* ≤ 0.00001	*p* ≤ 0.00001
Inoculation × Salt treatment	*p* ≤ 0.05	*p* ≤ 0.05	NS	NS

**Table 3 plants-09-01429-t003:** Effects of 100 mM NaCl and rhizospehere inoculation with *P. mandelii IB*-Ki14 and *B. subtilis IB*-22 on the shoot and root dry mass of 14-days-old wheat plants.

NaCl Concentration, mM	Bacterization	Dry Mass, mg	
		Root	Shoot
0	Without bacterization	14.2 ± 0.8 ^b^	40.0 ± 1.9 ^d^
100	Without bacterization	9.3 ± 0.7 ^a^	24.9 ± 0.6 ^a^
100	*P. mandelii* IB-Ki14	10.4 ± 0.9 ^a^	30.7 ± 0.5 ^b^
100	*B. subtilis* IB-22	10.8 ± 0.7 ^a^	34.3 ± 1.0 ^c^

Significantly different means for each variable are labeled with different letters (*p* ≤ 0.05; LSD test).

**Table 4 plants-09-01429-t004:** Effects of 100 mM NaCl and rhizosphere inoculation with *P. mandelii* IB-Ki14 and *B. subtilis* IB-22 on the transpiration (mg H_2_O h^−1^ plant^−1^), stomatal conductance (mmol m^−2^ s^−1^), and hydraulic conductivity (mg H_2_O m^-2^ s ^-1^ MPa^-1^) of 13-days-old wheat plants.

NaCl Concentration, mM	Treatment	Transpiration	Stomatal Conductance	Hydraulic Conductivity
0	Without bacteria	177 ± 10 ^c^	101 ± 9 ^b^	170 ± 21 ^b^
100	Without bacteria	108 ± 13 ^a^	77 ± 8 ^a^	77 ± 8 ^a^
100	*P. mandelii* IB-Ki14	134 ± 4 ^b^	65 ± 6 ^a^	64 ± 5 ^a^
100	*B. subtilis* IB-22	162 ± 5 ^c^	60 ± 5 ^a^	136 ± 15 ^b^

Significantly different means for each variable are labeled with different letters (*n* = 8, *p* ≤ 0.05; LSD test).

**Table 5 plants-09-01429-t005:** Analysis of variance of ABA content in roots and shoots of wheat plants untreated with bacteria or inoculated with either *B. subtilis* IB-22 or *P. mandelii* IB-Ki14 (inoculation) grown in the presence and absence of NaCl (salt treatment). *p* values are presented for effects of inoculation and salt-treatment and interaction of these factors (NS—non-significant).

Effect or Interaction	Root	Shoot
Salt treatment	*p* ≤ 0.0001	*p* ≤ 0.0001
Inoculation	*p* ≤ 0.05	*p* ≤ 0.05
Inoculation × salt treatment	*p* ≤ 0.05	*p* ≤ 0.05

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
