# Peer review of "Effects of Plant Growth Promoting Rhizobacteria on the Content of Abscisic Acid and Salt Resistance of Wheat Plants"

_plants, 2020, doi:10.3390/plants9111429_

Round 1
Reviewer 1 Report
The paper tackles an interesting observation of bacteria effect on stressed wheat ABA content in roots/shoots, including also field experiments. I find however some problems in experimental design.
Here are my questions
- Why use Elisa for ABA measurement, when there are readily available more accurate and newer methods (hplc-uv, mass spectrometry) ?
- How can we differentiate between, especially in roots, ABA synthesized by plant vs ABA from bacteria - which may be stuck to root/inproperly washed ? A description of roots & plant material collection and handling, from harvest till measurement , is missing from "Laboratory experiments/ABA assay" part
- Were the four plants, sampled for ABA analysis in lab experiments, taken from different vessels ( as one vessel contains ten seedlings ) ? How many biological replicates were used for one measurement ? Please add to "ABA assay" part
- Is it known if ABA synthesized by bacteria can enter plant through root ?
- Was the bacteria culture measured for ABA content before innoculation/after centrifugation ( described in last sentence of "Bacterial strains and culture media") ?
- How well is the innoculation with bacteria done - how is it ensured that, no/very little, nutrients from bacteria growth media are introduced to plants upon innoculation, thus affecting the plants ?
- Wouldnt a control experiment done through innoculation with clear bacterial media (the same media as used for bacteria, but without bacteria) be necessary ?
- How much of bacterial suspension in volume was, in total, added to one sample/vessel, in laboratory experiments ? Doesnt this volume contribute to diluting NaCL concentration ?
- Can we compare the field and lab experiments, regarding the different age/developmental stage of plants ? Is there a difference ?
- On pg. 9, line 334 please change "100 mm" to "100 mM"
- Are the four citations in ABA measurement method necessary ? Seems too much.
Author Response
We are most grateful to the reviewer for attentive reading of our article and useful comments. We have done our best to use all of them for improving our article. The comments of reviewer and our responses are as follows
- Why use Elisa for ABA measurement, when there are readily available more accurate and newer methods (hplc-uv, mass spectrometry)?
Response: Unfortunately HPLC combined with mass-spectrometry is not always available for us. But we have been able to check reliability of results of hormone immunoassay against HPLC-MS and whenever this was done, similar results were obtained with both methods. This was possibly due to modified method of purification of ABA and high specificity of antibodies we use. Description of purification of samples for hormone analysis was obviously too short and we added more information telling that (lines 422-432):
“In short ABA was partitioned with diethyl ether from the aqueous residue, after its dilution with distilled water and acidification with HCl to pH 2.5 (ratio of organic to aqueous phases being 3:1). Then, ABA was transferred from the organic phase into a solution of NaHCO3, (ratio of aqueous/organic phase being 1:3), reextracted from the acidified aqueous phase with diethyl ether, ABA quantitative determination was performed with ELISA using specific antibodies as described [38,39]. The reliability of the method was due to the specificity of the antibodies obtained against ABA and the use of a modified ABA extraction method based on reducing the volume of extractants at each stage of extraction and re-extraction, which allows efficient recovery of ABA while reducing the amount of extracted impurities. The sufficiency of ABA purification prior to immunoassay was proved by studying the chromatographic distribution of the immunoreactive material, which showed that the peak of immunoreactivity coincides only with the position of the internal ABA standard.”
2. How can we differentiate between, especially in roots, ABA synthesized by plant vs ABA from bacteria - which may be stuck to root/inproperly washed ? A description of roots & plant material collection and handling, from harvest till measurement , is missing from "Laboratory experiments/ABA assay" part
Response: The strains used in our experiments were characterized by very low level of ABA production (bacteria did not accumulate this hormone in the culture media to significant level). Still, reviewer was right and it was important to mention this. We have done it in the revised version by telling (lines 351-354) that “and were characterized by very low level of ABA accumulation measured in the culture media before inoculation (1-2 ng/ml, not less than 300 times lower than IAA and cytokinin concentrations in corresponding cytokinin (Bacillus subtilis IB-22 [33]) and auxin (Pseudomonas mandelii IB-Ki14 [34]) producing strains). “. Still we thoroughly washed the roots from the soil before extraction started and this is indicated in the present version (thanks to the reviewer) (line 417): “Before homogenization, roots were thoroughly washed from the soil”
3. Were the four plants, sampled for ABA analysis in lab experiments, taken from different vessels (as one vessel contains ten seedlings ) ? How many biological replicates were used for one measurement ? Please add to "ABA assay" part
Response: Sorry for writing this so unclearly. We mentioned the number of replicates in the figure legends, but the reviewer is right and this was not sufficient. So we rectified this by telling in the M&M section of in the revised version (417-418): “Shoots and roots of 4 plants were sampled from different vessels (number of replicates, n=6) (shoots of 4 plants and roots of 25 plants were sampled for each repetition (n=6) in the field experiments)”
4. Is it known if ABA synthesized by bacteria can enter plant through root?-
Response: The reviewer raised a very important question and we decided to address it by introducing a new paragraph into Discussion. It is as follows (lines 286-292)
“It was suggested that ABA released by microorganisms into the soil may contribute to an increase in the concentration of this hormone in the roots [26]. Addition of synthetic ABA into the nutrient solution did increase its concentration in the roots [21]. Although bacteria used in the present experiments did not accumulate ABA in vitro, this may change after their introduction into the soil. Nevertheless, ABA concentration in the shoots was decreased by bacterial inoculation suggesting that the changes in ABA concentration in the plants detected in the present experiments were not merely due to the uptake of bacterial ABA by the plants”
5. Was the bacteria culture measured for ABA content before innoculation/after centrifugation ( described in last sentence of "Bacterial strains and culture media") ?
Response: We have introduced this information according to the previous remark of the reviewer. It is now mentioned that ABA accumulation was measured in the culture media before inoculation (lines 352-355)
6. How well is the innoculation with bacteria done - how is it ensured that, no/very little, nutrients from bacteria growth media are introduced to plants upon innoculation, thus affecting the plants?
Response: This is a very useful comment, which prompted the necessary to clarify some experimental details. Thus we added See lines 389-391:” Since total volume of water in the pot was about 200 ml, addition of 10 ml (1 ml to each of ten seedlings in the pots) of either tap water or bacterial cells separated from the nutrient solution by centrifugation did not significantly influence concentrations of mineral nutrients.”
7. Wouldnt a control experiment done through innoculation with clear bacterial media (the same media as used for bacteria, but without bacteria) be necessary?
Response: Yes! This is important and has been previously checked. Sorry for not mentioning this in the original variant. It is mentioned now (line 394-395) that “Diluted supernatant obtained after peletting bacterial cells did not influence plants growth.”
8. How much of bacterial suspension in volume was, in total, added to one sample/vessel, in laboratory experiments ? Doesnt this volume contribute to diluting NaCL concentration ?
Response: Sorry for not mentioning that tap water was added to the unbacterized plant and thanks to reviewer for drawing our attention to this deficiency. This was rectified by adding that (lines 388-391) “Tap water (1 ml per each seedling) was added to the plants untreated with bacteria. Since total volume of water in the pot was about 200 ml, addition of 10 ml (1 ml to each of ten seedlings in the pots) of either tap water or bacterial cells separated from the nutrient solution by centrifugation did not significantly influence concentrations of NaCl.”
9. Can we compare the field and lab experiments, regarding the different age/developmental stage of plants ? Is there a difference ?
Response: This is interesting question and we added in accordance that (lines 238-241) “Although plants grown in laboratory and field experiments were sampled at different stages of development (the appearance of the third leaf in the case of laboratory experiments and the stage of tillering in the field experiments), the similarity of effects confirms that the effects persisted throughout development.”
10. On pg. 9, line 334 please change "100 mm" to "100 mM"
Response: this is the only point, where we dare to insist on our variant. It is usual for soil science to express the content of elements as millimoles per kilogram of soil (mmol/kg soil). See, e.g., Luciano et al. Numerical approach to modelling pulse-mode soil flushing on a Pb-contaminated soil. Journal of Soils and Sediments. 2013. 13, 43–55.
11. Are the four citations in ABA measurement method necessary ? Seems too much.
Response: We just wanted to confirm that results obtained by us with this method have been successfully published many times (the total number our article reporting on the results of ABA immunoassay reaches several dozen). But we can leave two of them, if reviewer thinks it important.
Reviewer 2 Report
During this work, two bacterial strains: Gram-positive aerobic cytokinin-producing bacterium Bacillus subtilis IB-22 and gram-negative auxin-producing bacterium Pseudomonas mandelii IB-Ki14 were inoculated into durum wheat seeds in order to test their effects on plant growth ABA content under salinity. Two experimental designs were conducted: laboratory and field experiments. Bacterial treatment increased ABA root content in contrast to un-inoculated plant, but bacterial treatment reduced ABA shoot content compared to un-inoculated control plants. Then the authors concluded that bacteria can raise salt resistance in wheat by increasing root ABA. However, more analyses and statistical tests are required in order to conclude the relationship between salinity stress, root and shoot growth, ABA content, ABA root to shoot ratio, and bacterial inoculation.
The major comments are:
- The experiments to test the effect of seed bacterization on crop yield under stress-free conditions and salinity was done for 2 years (2016-2019). Why the 2016 experiment was conducted under salinity-free conditions?
- There were huge variations in crop yields within 3 years under stress-free condition. For example, for IB-Ki14 inoculation and under no-stress, the yields were 876, 557, and 478 for 2016, 2018, and 2019, respectively. What the explanation of that large difference at the same conditions?
- I expected to see the effect of bacterial inoculation on dry and fresh weights of root and shoot system under salinity stress. Dry shoot and root masses are very significant indicators to determine the plant growth promoting activities of bacterial inoculation.
- In the Laboratory experiments, introduction of bacteria into the rhizosphere of salt-stressed plants increased the shoot weight and leaf area compared to plants grown in the presence of NaCl. Authors should clarify why refreshed bacterial inoculation into the rhizosphere was absent?
- Page 6, L 195, Authors mentioned that, “Evaluation of the effects of salinity and bacterial treatment on ABA concentration…..” This type of data should be analyzed using two-way ANOVA. The obtained data was analyzed using one-way ANOVA. So, the authors should clarify whether the experimental design was conducted to be analyzed using one OR two-way ANOVA?
- Page 7 (L227-229), Page 8 (L241-245) and Page 8 (L251-270). As we know linkage between ABA content and root or shoot growth is very critical point to ascertain the role of ABA in relation to salinity stress. In your experiment, there is another factor (bacterial inoculation) which actually affects that relationship. These kinds of data should be analyzed using another tests, not only using one way ANOVA.
Author Response
We are most grateful to the reviewer for attentive reading of our article and useful comments. We have done our best to use all of them for improving our article. The comments of reviewer and our responses are as follows:
- The experiments to test the effect of seed bacterization on crop yield under stress-free conditions and salinity was done for 2 years (2016-2019). Why the 2016 experiment was conducted under salinity-free conditions?
Response: We started with the field experiments under normal conditions without artificial salt-treatment of soil. Later on stress was applied to follow effects of PGPR on plant responses to salt treatment. According with the remark of the reviewer we added a sentence telling that (lines 85-86)“Later on salinization of the soil was applied to follow effects of PGPR on plant salt resistance.”
2. There were huge variations in crop yields within 3 years under stress-free condition. For example, for IB-Ki14 inoculation and under no-stress, the yields were 876, 557, and 478 for 2016, 2018, and 2019, respectively. What the explanation of that large difference at the same conditions?
Response: variation of crop yields is typical for our region, where two-fold changes in productivity are quite usual depending on variability of conditions (rainfall and temperature). To address this problem and in accordance with the remark of the reviewer we added (lines 79-82) that “Year-to-year variation of wheat crop yield was rather great, which was likely to be due to unstable temperature conditions in this region. Productivity was about 1.5 times higher in 2016 than in 2018, which may be explained by the fact that average temperature in May–June was 1.5 times higher in 2017 (17°C) than in 2018 (12°C)”.
3. I expected to see the effect of bacterial inoculation on dry and fresh weights of root and shoot system under salinity stress. Dry shoot and root masses are very significant indicators to determine the plant growth promoting activities of bacterial inoculation.
Response: In accordance with this valuable remark we added data on dry weight. In the revised version, the data are in the table 3. We also modified the text on the weight effects (lines 158-159) telling: “Salt-stress led to about 35 % decline in the accumulation of the both fresh and dry weight of shoots and leaf area compared to the stress-free plants (control) (Figure 4a and Table 3).” And (line 174-176) “The tendency of the increase in root dry mass resulting from inoculation with both strains was statistically insignificant (Table 3).”
4. In the Laboratory experiments, introduction of bacteria into the rhizosphere of salt-stressed plants increased the shoot weight and leaf area compared to plants grown in the presence of NaCl. Authors should clarify why refreshed bacterial inoculation into the rhizosphere was absent?
- Response: In the field experiments inoculation was preformed only once by bacterization of seeds and this was found to be sufficient to increase wheat yield. Persistence of effect of single inoculation was due to colonization of plants roots with the studied bacteria detected and published by us previously [34]. Accordingly we performed only single inoculation of seedlings without refreshed bacterial inoculation, which turned out to be sufficiently effective. Still, we found it reasonable to add some additional information in accordance with the remark of reviewer. It is now mentioned in the revised variant (lines 391-394) that “Only single bacterial inoculation was performed without refreshed bacterial inoculation into the rhizosphere, since the used bacteria have been previously shown to efficiently colonize roots and persist on their surface [34]”
5. Page 6, L 195, Authors mentioned that, “Evaluation of the effects of salinity and bacterial treatment on ABA concentration…..” This type of data should be analyzed using two-way ANOVA. The obtained data was analyzed using one-way ANOVA. So, the authors should clarify whether the experimental design was conducted to be analyzed using one OR two-way ANOVA?
Response: Sorry for poor description of statistics and thanks to the reviewer for useful advises. In the revised variant we tried to clarify this. It is now told that (lines 440-443) “Data were expressed as means ± S.E., which were calculated in all treatments using MS Excel. Significant differences between means were analysed by two-way analysis of variance (ANOVA) with bacterial and salt treatment as main factors, and a least significance difference (LSD) test to discriminate means.”
6. Page 7 (L227-229), Page 8 (L241-245) and Page 8 (L251-270). As we know linkage between ABA content and root or shoot growth is very critical point to ascertain the role of ABA in relation to salinity stress. In your experiment, there is another factor (bacterial inoculation) which actually affects that relationship. These kinds of data should be analyzed using another tests, not only using one way ANOVA.
Response: In accordance with the recommendation of the review we introduced tables with the results of two-way ANOVA and their description: (lines 147-151) “Analysis of variance of both fresh and dry mass of roots and shoots of wheat plants (Table 2) gave similar results: they showed significant effects of both inoculation and salt-treatment on the shoot mass, while only the effect of salt-treatment was significant in the roots. Nevertheless, the effect of salt-treatment on the root mass depended on inoculation supported by significant interaction between the factors (inoculation x salt treatment)” and (lines 216-219) “Two-way ANOVA showed significant effects of both salt-treatment and inoculation on ABA content in either roots or shoots. The effects of salt treatment on ABA level depended on inoculation in accordance with significant interaction between the factors (inoculation x salt treatment) revealed with the help of two-way ANOVA (Table 5).
We also added a phrase about ANOVA to the Discussion (After the sentence in the lines 227-229 (lines 261-265 in the present variant)):“Consequently, the decline in the level of stress-induced accumulation of ABA detected in inoculated plants could contribute to the activation of plant growth by maintaining chlorophyll level and photosynthesis. Dependence of ABA content in the shoots on inoculation was in accordance with significant interaction between the factors (inoculation x salt treatment) revealed with the help of two-way ANOVA (Table 5).”
After the phrase (lines 214-245, now 277-282): “Consequently, the increase in the level of this hormone in plant roots detected in our experiments, which is most clearly manifested in plants inoculated with B. subtilis IB-22, could promote the activation of root growth, causing an increase in their ability for water uptake” we added a sentence concerning the effect of inoculation: “Dependence of ABA content in the shoots on inoculation was in accordance with significant interaction between the factors (inoculation x salt treatment) revealed with the help of two-way ANOVA (Table 5).”
We hope that these references to Table 5 may be sufficient to support effects of inoculation on ABA content in shoots and roots and allow further discussion of importance of the effects of inoculation on ABA content.
Round 2
Reviewer 1 Report
The reviewer would like to thank the authors for the swift and appropriate response, with good addressing of most of the points.
I would however require just two final things
- Referring to point 11, please leave two citations.
- Please send or show data for the experiment of inoculation, with clear bacterial media without bacteria, as referred in point 7. This experiment shouldn't be done with "pelleted bacteria", just with clear media. This experiment should contain everything except bacteria.
Author Response
We are most grateful to reviewer for the kind words about our revised version and calling our changes “appropriate”. This was mostly due to detailed analysis of our work by reviewer.
In accordance with two final things required by reviewer
- We left only two citations concerning ABA purification and analysis.
- In accordance with the recommendation to show data for the experiment of inoculation, with clear bacterial media without bacteria we modified the sentence (lines 398-400) by telling “Diluted supernatant obtained after peletting bacterial cells as well as the clear nutrient media processed in the same way as bacterial preparation did not influence plants growth.”
Since reviewers were satisfied with the changes in the previous revised version of our manuscript those changes were accepted and are not tracked in the present version. Only changes made according to the minor revision are tracked
Reviewer 2 Report
All my comments were addressed in the revised version.
Just, Table 3 should be correctely layout
Author Response
We are happy that we managed to satisfy the reviewer by addressing most of comments. Table 3 is correctly layout in the present version. Since reviewers were satisfied with the changes in the previous revised version of our manuscript those changes were accepted and are not tracked in the present version. Only changes made according to the minor revision are tracked